# Electrolyte design principles for developing quasi-solid-state rechargeable halide-ion batteries

Xu Yang[1,2], Bao Zhang [3,4], Yao Tian[2], Yao Wang[2], Zhiqiang Fu[2], Dong Zhou [2] ✉, Hao Liu [1], Feiyu Kang [2], Baohua Li [2] ✉, Chunsheng Wang [3] ✉ & Guoxiu Wang [1] ✉

Rechargeable halide-ion batteries (HIBs) are good candidates for large-scale due to their appealing energy density, low cost, and dendrite-free features. However, state-of-the-art electrolytes limit the HIBs' performance and cycle life. Here, via experimental measurements and modelling approach, we demonstrate that the dissolutions in the electrolyte of transition metal and elemental halogen from the positive electrode and discharge products from the negative electrode cause the HIBs failure. To circumvent these issues, we propose the combination of fluorinated low-polarity solvents with a gelation treatment to prevent dissolutions at the interphase, thus, improving the HIBs' performance. Using this approach, we develop a quasi-solid-state Cl-ion-conducting gel polymer electrolyte. This electrolyte is tested in a single-layer pouch cell configuration with an iron oxychloride-based positive electrode and a lithium metal negative electrode at 25 °C and 125 mA g$^{-1}$. The pouch delivers an initial discharge capacity of 210 mAh g$^{-1}$ and a discharge capacity retention of almost 80% after 100 cycles. We also report assembly and testing of fluoride-ion and bromide-ion cells using quasi-solid-state halide-ion-conducting gel polymer electrolyte.

Lithium (Li)–ion batteries have dominated the market of portable energy storage devices in the past few decades due to their high energy density. However, they are not economically viable to meet the ever-increasing demands of large-scale stationary energy storage due to the high cost[1] of raw materials. Therefore, the exploitation of new rechargeable battery systems based on more abundant elements is in urgent need. Recently, non-Li metal-ion batteries, including alkali metal-ion batteries (e.g., sodium (Na)–ion and potassium (K)–ion batteries[2]) and multivalent metal-ion batteries (e.g., calcium (Ca)–ion, magnesium (Mg)–ion, aluminum (Al)–ion, and zinc (Zn)–ion batteries[3]), have attracted extensive attention and been studied as cost-effective alternatives of Li-ion batteries. However, it remains a

challenge to develop non-Li metal-ion batteries with long cycle life and low cost without sacrificing energy density.

Halide-ion batteries (HIBs; e.g., fluoride (F)–ion, chloride (Cl)–ion, and bromide (Br)–ion batteries) with anion species as charge carriers are competitive for low-cost energy storage due to their relatively high energy/power densities (>500 Wh L$^{-1}$) and the resource abundance of halogen elements[1,4,5]. In analogy to Li-ion batteries, HIBs follow a "rocking-chair" chemistry in which halide ions (F$^-$, Cl$^-$, and Br$^-$) shuttle between the halide-ion storage cathodes (metal halides[6], metal oxyhalides[7], organic halides[8], etc.) and highly electropositive metal anodes (Li[9], Mg[10], etc.). For example, Zhao et al. developed metal chlorides[6] and metal oxychloride[9] as the reversible electrode materials

[1]Centre for Clean Energy Technology, School of Mathematical and Physical Sciences, Faculty of Science, University of Technology Sydney, Sydney, NSW 2007, Australia. [2]Tsinghua Shenzhen International Graduate School, Tsinghua University, Shenzhen 518055, China. [3]Department of Chemical and Biomolecular Engineering, University of Maryland, College Park, MD 20742, USA. [4]School of Optical and Electronic Information, Huazhong University of Science and Technology, Wuhan 430074, China. ✉e-mail: zhou.d@sz.tsinghua.edu.cn; libh@sz.tsinghua.edu.cn; cswang@umd.edu; Guoxiu.Wang@uts.edu.au

for liquid Cl-ion batteries (CIBs) in 2013; Davis et al. reported an F-ion-conducting liquid electrolyte to support the room-temperature operation of rechargeable F-ion batteries (FIBs) in 2018[11]. FIBs and CIBs can achieve high theoretical volumetric energies up to 5000 Wh L$^{-1}$ and 2500 Wh L$^{-1}$ (Fig. 1a and Supplementary Table 1), which are comparable to that of Li-oxygen and Li-sulfur batteries, respectively[9,11]. Furthermore, the single electron transfer reaction mechanism provides HIBs with faster kinetics of electrode reaction than those of multivalent metal-ion batteries[1]. Besides, HIBs are intrinsically dendrite-free on the metallic anodes, thus eliminating the safety issues (e.g., short circuit and thermal runaway) which generally exist in metallic anode-based cation-shuttle batteries[12].

Despite the above merits, the research progress on HIBs is far from satisfactory. The current state-of-the-art HIBs generally suffer

from poor compatibility between electrodes and the halide-ion-conducting electrolytes (including aqueous electrolytes[13], non-aqueous liquid electrolytes[12], and solid-state electrolytes[14]). The narrow electrochemical window of aqueous electrolytes strongly limits their battery energy density, while the low ionic conductivity and the poor interfacial contact of solid-state electrolytes hinder the room-temperature operation of HIBs[15]. With regard to the non-aqueous liquid electrolytes, in which halide salts are dissolved in carbonate[7,12,16–18] or ionic liquid[8–10,19–24] solvents, the cathodes suffer from severe structural deterioration along cycling, while the possible failure mechanism, including volume changes, electrode delamination[6,19,20] and Lewis base-derived cathode corrosion[8–10], remains controversial. Moreover, the compatibility between metallic anodes and electrolytes, especially the construction of halide-ion-

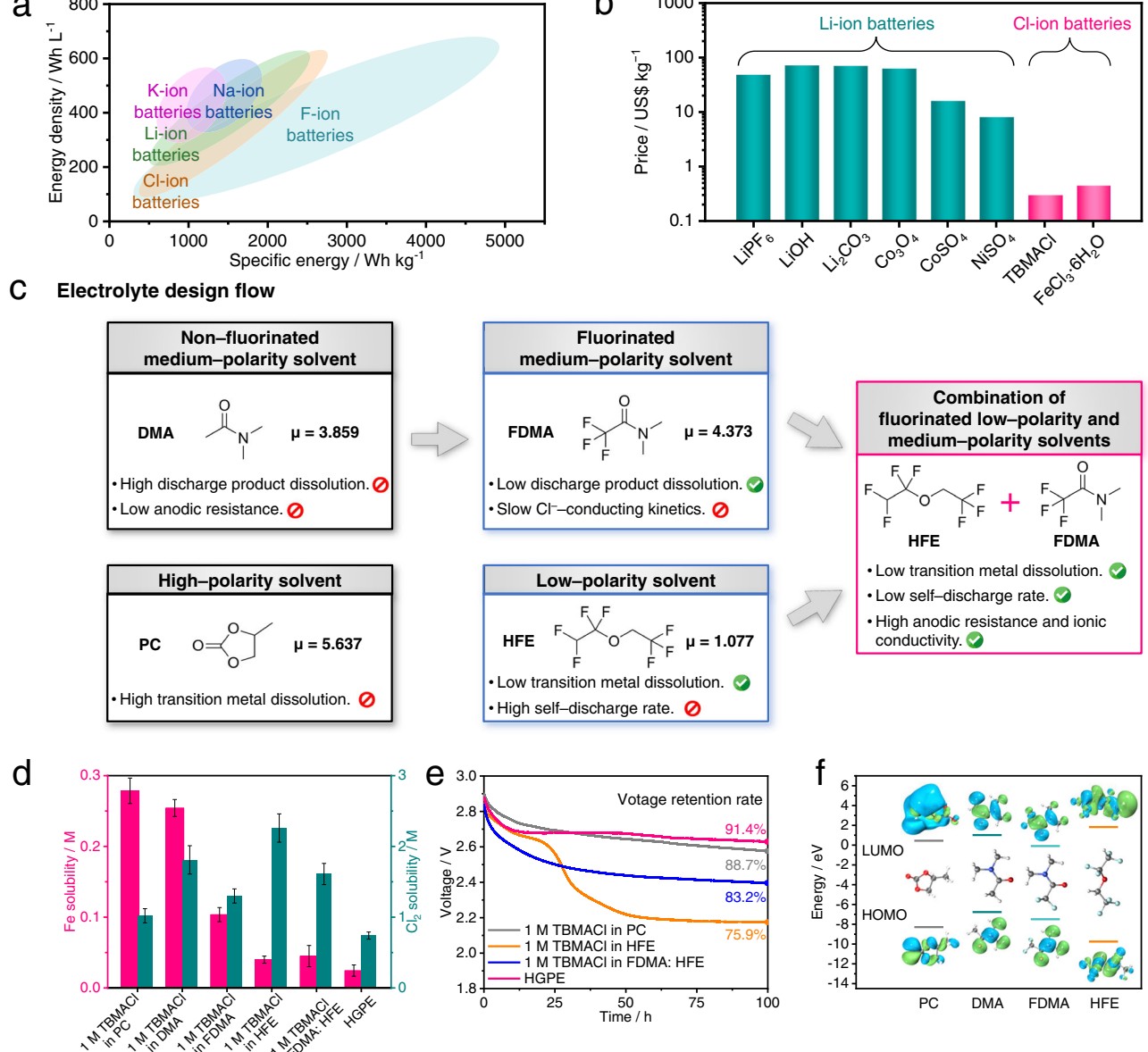

**Fig. 1 | Electrolyte screening for CIBs. a** Comparison of the theoretical energy density and specific energy of Li−, Na−, K−, F−, and Cl-ion batteries. The theoretical specific energies are calculated based on the total mass of the anode and cathode active materials. **b** Price of raw materials for Li− and Cl-ion batteries. **c** Schematic representation of the molecular strategy used for the development of CIB binary non-aqueous electrolyte solvents. μ is the dipole moment. **d** Fe and Cl$_2$ solubility in different electrolyte samples at 25 °C. Error bars represent standard error of the mean. **e** Self-discharge profiles of fully charged Li‖FeOCl coin cells with different electrolytes at 25 °C. **f** LUMO and HOMO energy values of PC, DMA, FDMA, and HFE molecules. The corresponding molecular structures and visual geometry structures are shown in the insets. Grey, white, red, blue, and cyan balls represent carbon, hydrogen, oxygen, nitrogen, and fluorine atoms, respectively.

conducting passive interphase[19], has been rarely considered. All these aspects limit the reversible capacity and cycle life (<100 cycles) of current non-aqueous liquid electrolyte-based HIBs. Therefore, screening suitable electrolyte solvents to support high-performance HIBs remains an utmost challenge.

Herein, we report the design principle of non-aqueous electrolytes to enhance the performance of HIBs. We demonstrate that the failure of HIBs is dominated by the dissolution of both cathode transition metal, elemental halogen, and discharge products in liquid electrolyte solutions, which can be efficiently suppressed by a combination of fluorinated less-polar solvents. In situ gelation of the fluorinated solvent-based electrolyte hinders the dissolutions at the interphase, thus, improving the battery performance. As a proof of concept, we show a rationally designed quasi-solid-state halide-ion-conducting gel polymer electrolyte (HGPE) which can enhance the reversible capacity and cyclability of CIBs. The HGPEs were prepared via in situ polymerizations of a liquid electrolyte composed of tributyl methyl ammonium chloride (TBMACl) salt dissolved in a mixture of 2,2,2−trifluoro−N, N-dimethylacetamide (FDMA) and 1,1,2,2−tetrafluoroethyl 2,2,2−trifluoroethyl ether (HFE) solvents. It shows adequate ionic conductivity ($2.64\,mScm^{-1}$ at 25 °C), high oxidation stability (4.0 V vs. Li$^+$/Li), and safety aspects (i.e., non-flammability and leakage-free). When the HGPE is tested in combination with a 50 μm-thick Li metal negative electrode and FeOCl-based positive electrode, the CIB delivers a high initial discharge capacity of $230\,mAhg^{-1}$ at 25 °C and $125\,mA\,g^{-1}$ with a stable cyclability (capacity retentions of 85.2% after 250 cycles and 68.3% after 500 cycles). Theoretical modeling and experimental investigations reveal that the side reactions occurring at the interphase could be restrained by the tailored solvation structure and polymer matrix of the HGPE. It is worth noting that this CIB is mechanistically different from the recently reported Na (Li)/Cl$_2$ batteries system[25], in which alkali metal ions as charge carriers shuttle between the electrodes, and the dendrites growth concerns are still present for the alkali metal electrodes. Furthermore, we demonstrate that the HGPE concept can also be extended to produce reversible FIBs and Br-ion batteries (BIBs), which opens a promising development pathway for secondary halide-ion batteries.

## Results and discussion
### Electrolyte design for HIBs
To demonstrate the requirements of electrolytes for HIBs, the CIBs based on Li∥FeOCl cell are chosen as a representative model, based on reactions as follow[9]:

$$Cathode : FeOCl + e^- \leftrightarrow FeO + Cl^- \tag{1}$$

$$Anode : Li + Cl^- \leftrightarrow LiCl + e^- \tag{2}$$

As seen from Fig. 1b and Supplementary Table 2, the costs of the cathode raw materials and electrolyte salt in the CIBs are lower than those in Li-ion batteries.

Three side reactions related to the dissolution effect are present in liquid electrolytes for CIBs (Supplementary Table 3). The first side-reaction is the dissolution of transition metal:

$$3\,FeOCl \rightarrow Fe_2O_3 + Fe^{3+} + 3\,Cl^- \tag{3}$$

Second, the self-discharge reaction leads to the dissolution of elemental chlorine (Cl•):

$$FeOCl \rightarrow FeO + Cl· \tag{4}$$

The third side-reaction is the dissolution of discharge product (i.e., LiCl):

$$LiCl \rightarrow Li^+ + Cl^- \tag{5}$$

Consequently, an ideal electrolyte for CIBs should not only demonstrate high ionic conductivity to enable the rapid Cl$^-$−conducting kinetics and wide electrochemical stability window to suppress the anodic decomposition, but also exhibit low solubility of Fe element, elemental Cl, and LiCl.

Therefore, to screen suitable electrolyte solvent to meet the above requirements, the solvent candidates with different dipole moments (μ, which were obtained based on density functional theory (DFT) calculation[26]) are classified into high-polarity solvents (including propylene carbonate (PC) with a μ = 5.637), medium-polarity solvents (including N, N-dimethylacetamide (DMA) with a μ = 3.869 and FDMA with a μ = 4.373) and low-polarity solvents (including HFE with a μ = 1.077, Fig. 1c). 1 M TBMACl was dissolved in above solvents for electrolyte preparation. First, it is seen that the solubility of the Fe element from the FeOCl cathode in these electrolytes increase with solvent polarity (Fig. 1d). Therefore, in the high-polarity solvent (i.e., PC)−based electrolytes (the lower left panel of Fig. 1c), the high transition metal dissolution (Eq. 3) leads to irreversible chemical deterioration of the cathode material (Supplementary Fig. 1 and Supplementary Table 4). Furthermore, it is seen from Fig. 1d and Supplementary Fig. 2 that the solubility of Cl$_2$ increases with the reduced solvent polarity. Consequently, in the low-polarity solvent (i.e., HFE)−based electrolytes (the lower middle panel of Fig. 1c), the FeOCl cathode suffers from a dissolution of elemental chlorine (Eq. 4). The soluble elemental chlorine continuously shuttles to the Li anode in the Li∥FeOCl cell, triggering cell self-discharge. This is verified by the open circuit voltage result in Fig. 1e, in which the voltage retention of fully charged Li∥FeOCl cell using 1 M TBMACl in HFE electrolyte is 75.9% of the initial value after aging for 100 h, which is lower than the cell using PC-based electrolyte (88.7%).

Medium-polarity solvents (e.g., DMA) were selected to reach a balance between the chemical stability of the cathode material and low self-discharge. Unfortunately, the solubility of discharge product (i.e., LiCl) in DMA solvent is as high as 1.7 M (Supplementary Fig. 3, corresponding to Eq. 4). This suggests an adverse solid-phase deposition of the discharge product on the Li metal anode in the presence of a DMA-based electrolyte (the upper left panel of Fig. 1c). The dissolved Li ions subsequently shuttle to the cathode surface, triggering electrochemical degradation on the electrode surface. Additionally, as shown in Supplementary Fig. 4, the anodic stability of 1 M TBMACl in DMA electrolyte is as low as 2.8 V vs. Li$^+$/Li, giving rise to the irreversible oxidative decomposition of electrolyte during the charging process.

To address the above issues, the −CH$_3$ on the DMA molecule was replaced by −CF$_3$ to form FDMA. As shown in Fig. 1f, after fluorination, the lower highest occupied molecular orbital (HOMO) energy of DMA (−6.768 eV) significantly reduced to −7.490 eV, corresponding to an extended oxidative threshold of the 1 M TBMACl in FDMA electrolyte (3.3 V vs. Li$^+$/Li, Supplementary Fig. 4). This validates that the electron-withdrawing effect of fluorine atoms can effectively extend the anodic stability of solvents. Furthermore, the FDMA solvent delivers a negligible LiCl solubility (0.01 M at 25 °C, Supplementary Fig. 3) compared with that of the DMA, thus, hindering the dissolution of the discharge product. However, the weak dissociation of TBMACl salt in FDMA-based electrolyte results in sluggish Cl$^-$ transport, which limits the kinetics of electrode reactions in CIBs (the upper middle panel of Fig. 1c). Accordingly, the medium-polarity FDMA is paired with the low-polarity HFE solvent in a volume ratio of 1: 1 to further optimize the electrolyte solvation structure. Besides the drawback of elemental

chlorine solubility as illustrated above, the HFE solvent possesses various merits including the lowest HOMO value among solvent candidates (−9.747 eV, Fig. 1f) associated with high anodic stability (Supplementary Fig. 4), low LiCl solubility (<1 μM at 25 °C) and high TBMACl solubility (Supplementary Fig. 3). As a result, the introduction of HFE to the 1 M TBMACl in FDMA electrolyte not only effectively decreases the LiCl solubility (<1 μM at 25 °C, Supplementary Fig. 3) and extends the electrochemical window to 3.7 V *vs.* Li⁺/Li (Supplementary Fig. 4), but also promotes the dissociation of TBMACl salt and facilitates the Cl⁻−conducting kinetics in the electrolyte (the right panel of Fig. 1c). Moreover, the combination of fluorinated less-polar solvents (i.e., FDMA and HFE) enhances the Li anode|electrolyte compatibility by forming a robust LiF-rich solid electrolyte interphase (SEI) on the Li metal surface to protect the electrode from undesired volume changes during the cell cycling.

On this basis, 1.5 wt% pentaerythritol tetraacrylate (PETEA) monomer was thermally polymerized in situ to gel the 1 M TBMACl in FDMA: HFE electrolyte into HGPE (Supplementary Fig. 5 and Supplementary Note 1). The gel matrix of HGPE eliminates the liquid-leakage issues (Supplementary Fig. 6 and Supplementary Note 2) after gelation. More importantly, the polymer matrix further hinders the dissolution of transition metal and elemental chlorine from the cathode (Fig. 1d), thus improving electrode|electrolyte interphase compatibility. The as-formed gel polymer electrolyte exhibits high oxidation stability up to 4.0 V *vs.* Li⁺/Li (Supplementary Fig. 4) and high ionic conductivity of 2.64 mScm⁻¹ at 25 °C, which is close to the pristine liquid electrolyte (2.76 mS cm⁻¹ at 25 °C, Supplementary Fig. 7 and Supplementary Table 5). Furthermore, this gel polymer electrolyte exhibits a self-extinguishing time (SET) of 0 s g⁻¹ (Supplementary Fig. 8 and Supplementary Movie 1, 2) due to the thermal stability and non-flammability feature of fluorinated solvents confined by polymer matrix[27].

## Electrochemical performance of Li|Cl-ion-conducting HGPE| FeOCl cells

The performance of Cl-ion-conducting HGPE-based CIBs was evaluated by coin cells with a FeOCl-based positive electrode (the FeOCl mass loading was around1.0 mg cm⁻²) and a 50 μm-thick Li metal negative electrode. The cyclic voltammetry (CV) curves of Li | |FeOCl cells with different electrolytes at 0.1 mV s⁻¹ are shown in Fig. 2a and Supplementary Fig. 9. It is seen that two oxidation peaks centralized at around 2.22 and 3.12 V with four corresponding reduction peaks at around 1.99, 2.17, 2.69, and 3.10 V appear in the CV curves, suggesting a multistep Cl-ion discharge/charge process of FeOCl cathode[20]. As shown in the CV curves of the Li|1 M TBMACl in the PC|FeOCl cell (Supplementary Fig. 9a), a pair of redox peaks around 3.1–3.2 V appears after the first cycle of CV curves, representing the $Fe^{3+}/Fe^{2+}$ redox owing to the high dissolution of $Fe^{3+}$ in PC-based electrolyte. The CV curves of Li|1 M TBMACl in the HFE|FeOCl cell displays a dominant anodic peak below 2.5 V, indicating cell self-discharge (Supplementary Fig. 9b). Meanwhile, the initial CV curve of the CIB with 1 M TBMACl in DMA electrolyte shows a strong oxidation current starting from 2.8 V (Supplementary Fig. 9c), suggesting a continuous oxidative decomposition of DMA solvent. The intensity of Cl⁻ redox reaction peaks rapidly decreases in the following cycles, which reflects the irreversible electrochemical behaviour of the FeOCl cathode. Moreover, the current intensities related to the above side reactions were decreased while the reversibility of CV curves was improved with the fluorination of DMA into FDMA as solvent (Supplementary Fig. 9d) and the introduction of HFE co-solvent (Supplementary Fig. 9e). The Li|HGPE|FeOCl cell exhibits well-overlapped CV curves with low polarization and negligible electrolyte oxidation (Fig. 2a), demonstrating the reversibility of this CIB system.

Li||FeOCl cells with various electrolytes were galvanostatically cycled at 125 mA g⁻¹ between 1.6 and 3.3 V, and their cycling performances are shown in Fig. 2b. The Li|1 M TBMACl in PC|FeOCl

(Supplementary Fig. 10a) and Li|1 M TBMACl in HFE|FeOCl (Supplementary Fig. 10b) cells deliver initial discharge capacities of 230 and 246 mAh g⁻¹ at 25 °C, respectively. Meanwhile, battery failures occurred at the 8th and 24th cycle, respectively. These results suggest the use of solvents with the highest and lowest polarities hinder their application as the mono solvent for Cl-ion-conducting electrolytes. The cell with 1 M TBMACl in medium-polarity DMA electrolyte suffers from low initial discharge capacity (168 mAh g⁻¹) with a Coulombic efficiency of 148.8% caused by the electrolyte oxidative decomposition (Supplementary Fig. 10c and Fig. 2b), and the capacity rapidly fades to about 0 mAh g⁻¹ after 6 cycles. However, the initial discharge capacities were improved, and the lifespans were extended in the Li|1 M TBMACl in FDMA|FeOCl (Supplementary Fig. 10d and Supplementary Fig. 11) and Li|1 M TBMACl in FDMA: HFE | FeOCl (Supplementary Fig. 10e and Supplementary Fig. 11) cells. Moreover, the Li|HGPE|FeOCl cell shows a high initial discharge capacity of 230 mAh g⁻¹ with a Coulombic efficiency of 97.4% (Fig. 2c) and maintains capacity retentions of 85.2% after 250 cycles and 68.3% after 500 cycles (Fig. 2b). This result is further corroborated by the Nyquist plots of EIS measurements (Supplementary Note 3) for Li||FeOCl coin cells with different electrolytes after 3 cycles (left inset of Fig. 2d) and the selected cycles (right inset of Fig. 2d). The EIS curves were fitted using an equivalent circuit (Supplementary Fig. 12 and Supplementary Table 6). The fitting result indicated that charge-transfer resistance ($R_{ct}$) and the interphase resistance ($R_f$) of the HGPE-based cell changes with smaller variations during cycling, compared with cells using other electrolytes (Fig. 2d). This implies good compatibility between HGPE and electrodes that contributes to the improved cycling performance in Fig. 2b. Indeed, it is possible to calculate a specific energy of about 432 Wh k⁻¹ (based only on the mass of FeOCl in the positive electrode and a cell discharge voltage of 2.4 V at 125 mA g⁻¹) for the quasi-solid-state Li|HGPE|FeOCl coin cell (Supplementary Note 4). This specific energy value is well-positioned in the literature for CIBs with organic-solvent-based electrolytes[7,12,15–18], ionic-liquid-based electrolytes[8–10,19–24], and aqueous electrolytes[13,28–30] (Supplementary Fig. 13 and Supplementary Table 7). Moreover, the Li|HGPE|FeOCl cell delivers discharge capacities of 243, 202, 140, and 85 mAh g⁻¹ at the specific current of 50 mA g⁻¹, 125 mA g⁻¹, 250 mA g⁻¹ and 500 mA g⁻¹, respectively, which are higher than those of the cells using mono solvent-based electrolytes and fluorinated liquid electrolytes (Fig. 2e, Supplementary Fig. 14a and Supplementary Fig. 14b), suggesting an improved Cl⁻ kinetics through the HGPE and electrode|electrolyte interphase. Based on the above results, the properties of electrolytes for conventional CIBs and HGPE in this work were compared (Supplementary Fig. 15). Except for the negligible decrease in ionic conductivity, our HGPE presents attractive functionalities over the conventional electrolytes for CIBs, including improved electrode compatibility, wide potential window, thorough non−flammability, non−leakage, and longer cycle life.

## Solvation structure of the Cl-ion-conducting electrolytes

Molecular dynamics (MD) simulations were carried out to understand the solvation structures of electrolytes (Supplementary Fig. 16a–d). As seen from the radial distribution function (RDF) of TBMA⁺−coordination pairs, the 1 M TBMACl in FDMA electrolyte exhibits a strong $N_{TBMA}$−Cl peak at 4.9 Å, suggesting a predominant existence of TBMA⁺-Cl⁻ clusters (Fig. 3a). It is seen that the intensity of this peak decreases after the introduction of HFE (Fig. 3b) associated with an increased peak intensity of $N_{TBMA}$−$O_{FDMA}$ pairs (Fig. 3a, b) and a reduced peak intensity of Cl−$H_{FDMA}$ pairs (Supplementary Fig. 17a, b). This indicates that the dissociation degree of TBMACl salt is effectively enhanced in 1 M TBMACl in FDMA: HFE electrolyte. Interestingly, compared with 1 M TBMACl in HFE electrolyte, the peak intensity of $N_{TBMA}$−$O_{HFE}$ pairs (Fig. 3b and Supplementary Fig. 17c) is weaker while the peak intensity of Cl−$H_{HFE}$ pairs is stronger (Supplementary

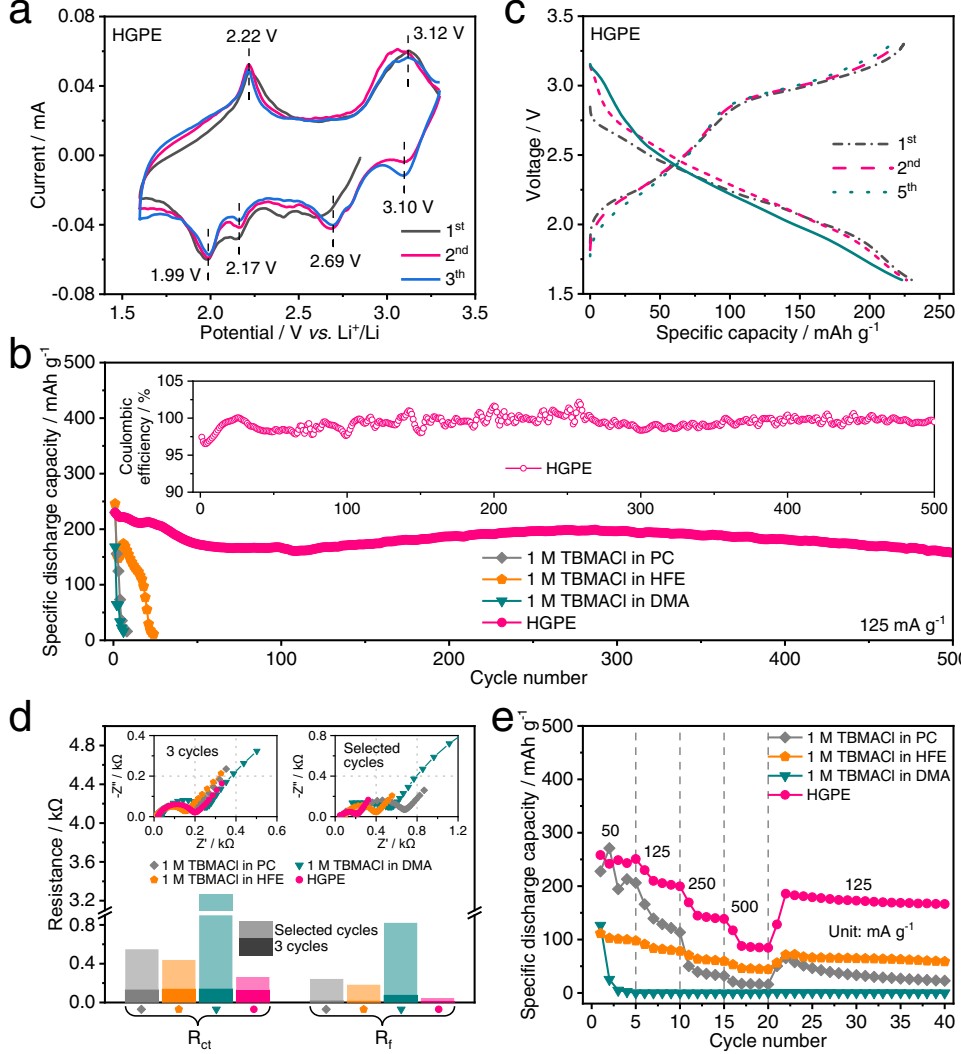

**Fig. 2 | Electrochemical performance of the quasi-solid-state CIBs. a** CV curves of the Li|Cl-ion-conducting HGPE|FeOCl coin cell at 0.1 mV s⁻¹. **b** Cycling performance of the Li||FeOCl coin cells with 1 M TBMACl in PC, 1 M TBMACl in HFE, 1 M TBMACl in DMA electrolyte and HGPE at 125 mA g⁻¹ and 25 °C. Inset is the Coulombic efficiency. **c** Typical galvanostatic discharge/charge profiles of the Li||FeOCl coin cells with HGPE at 125 mA g⁻¹ and 25 °C. **d** Nyquist plots of EIS measurements for Li||FeOCl coin cells with different electrolytes after 3 cycles and the selected cycles (left inset; 8, 24, and 6 cycles for 1 M TBMACl in PC, 1 M TBMACl in HFE, 1 M TBMACl in DMA electrolyte-based cells and 500 cycles for the HGPE-based cell, respectively) at 125 mA g⁻¹ and 25 °C. Right inset is the evolutions of $R_{ct}$ and $R_f$ of the CIBs after 3 cycles and the selected cycles. **e** Rate performance of the Li||FeOCl coin cells with different electrolytes.

Fig. 17b, d) in 1 M TBMACl in FDMA: HFE electrolyte. This can possibly be explained as follows. As shown in electrostatic potential (ESP) distribution obtained from DFT calculations in Fig. 3c, the negative charge distribution of FDMA molecules is more concentrated than that of HFE; on the contrary, the positive charge distribution of HFE molecules is more concentrated than that of FDMA. This reveals that the FDMA molecules prefer to solvate cations, while the HFE molecules tend to solvate anions. This different solvation inclination gives rise to a higher dissociation degree of TBMACl salt in binary solvents than that in a mono solvent via forming more TBMA⁺–FDMA and Cl⁻–HFE coordination structures. This synergistic effect can be verified by the fact that the bulk ionic conductivity of 1 M TBMACl in FDMA: HFE electrolyte (2.76 mS cm⁻¹ at 25 °C) is higher than those of 1 M TBMACl in FDMA (2.46 mS cm⁻¹ at 25 °C) and 1 M TBMACl in HFE (2.01 mS cm⁻¹ at 25 °C) electrolytes (Supplementary Fig. 7 and Supplementary Table 5), which is expected to promote the transport kinetics of Cl ions in interphase reactions. Notably, the solvation structure can be well maintained after the in situ gelations of 1 M TBMACl in FDMA: HFE electrolyte (Supplementary Fig. 17e, f).

Experimental evidence of the above results is obtained from the analysis of the Raman spectroscopy measurements of the electrolytes reported in Fig. 3d and Supplementary Fig. 18a. The stretching vibration of C=O in FDMA molecule creates a broad peak that can be further fitted to two subpeaks at 1709 and 1691 cm⁻¹, corresponding to free FDMA molecules and FDMA clusters linked by C=O···H–C bonds, respectively[31]. It is seen that after dissolving 1 M TBMACl salt into FDMA, a new peak appeared at about 1700 cm⁻¹ while the peak intensities of free FDMA and FDMA cluster significantly reduced, indicating the formation of solvation structure in which FDMA molecules coordinate with central TBMA⁺ cations[32]. Moreover, it is seen that after the introduction of HFE solvent, the peak intensity of the TBMA⁺–FDMA coordination structure dramatically increased (Fig. 3d). Meanwhile the intensity of asymmetric C–H stretching peak from Cl⁻–HFE clusters at 2895 cm⁻¹ is much higher compared with that in 1 M TBMACl in HFE electrolyte[33] (Supplementary Fig. 18b). This confirms the high dissociation degree of TBMACl salt in 1 M TBMACl in FDMA: HFE electrolyte, which is well consistent with the MD simulation in Fig. 3a, b.

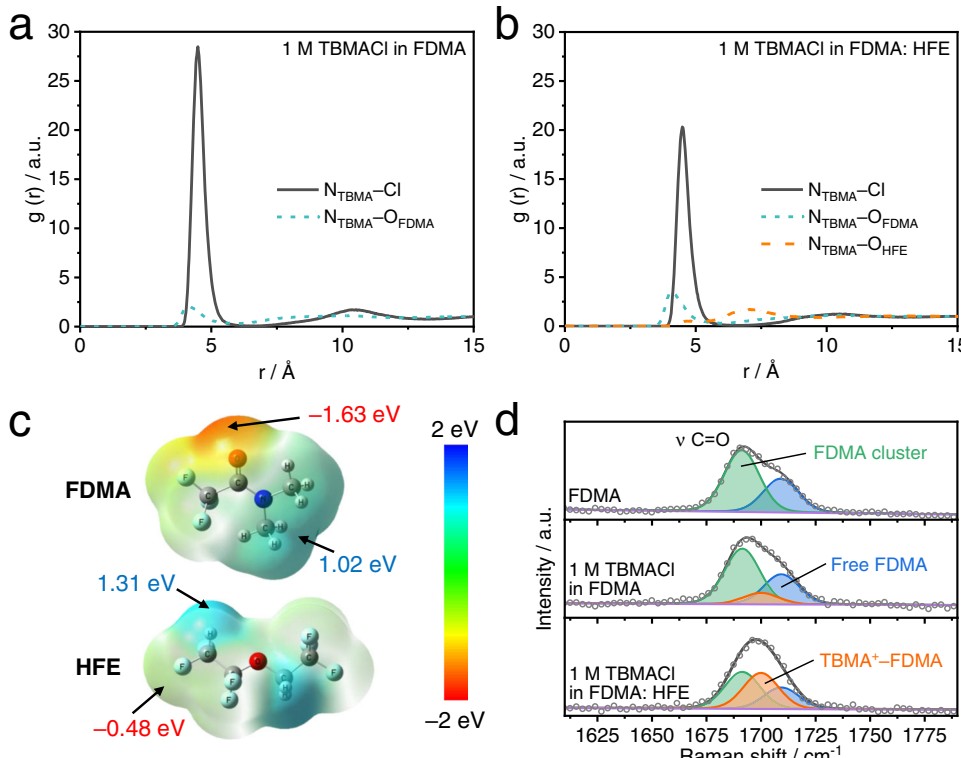

**Fig. 3 | Solvation structures of electrolytes. a, b** RDFs of TBMA$^+$–coordination pairs obtained from MD simulations for **a** 1 M TBMACl in FDMA and **b** 1 M TBMACl in FDMA: HFE electrolytes. **c** ESP distributions of FDMA and HFE molecules. **d** Raman spectra of FDMA solvent and different electrolytes at 25 °C.

## Physicochemical characterizations of the electrode|electrolyte interphases

The composition and morphology of different electrolyte|positive electrode interphases were investigated to unlock the effect of electrolyte components on interphase structure and chemical composition. As seen from the ex situ transmission electron microscopy (TEM) image (Supplementary Fig. 19a), an 8 nm-thick layer was formed on the FeOCl cathode surface (Supplementary Fig. 19b) after 3 cycles in 1 M TBMACl in a high-polarity PC electrolyte. Such layer mainly consists of Fe$_3$O$_4$ (PDF#01-089-6466) the reaction product of cathode dissolution and discharge process, which can be further confirmed by the ex situ in-depth X-ray photoelectron spectroscopy (XPS) electrode measurements and analyses in Supplementary Fig. 20a, b and Supplementary Note 5. For comparison, the of FeOCl (110) lattice fringes largely deteriorated at 3.3 V after 3 cycles at 25 °C and 125 mA g$^{-1}$ in 1 M TBMACl in low-polarity HFE electrolyte (Supplementary Fig. 21). The cycled positive electrode exhibits a structure composed of crystalline FeOCl regions mixed with amorphous FeO as the self-discharge product (Supplementary Fig. 22 and Supplementary Fig. 23a, b). The ex situ X-ray diffraction (XRD) patterns were shown in Supplementary Fig. 24a, b. It is worth noting that the organic cation of the ionic liquid solvent could insert into the layered FeOCl structure during the resting process leading to the interlayer expansion of FeOCl, reflected by the shift of (010) diffraction in the XRD pattern (Supplementary Fig. 25 and Supplementary Note 6). Furthermore, the structural reversibility of the FeOCl (010) peak in the HFE-based electrolyte is less than that in the PC-based electrolyte after a discharge/charge process, further supporting the hypothesis of the cell self-discharge in the low-polarity solvent-based electrolyte.

Figure 4a displays that after 3 cycles at 25 °C and 125 mA g$^{-1}$ in the 1 M TBMACl in medium-polarity DMA electrolyte, a 6 nm-thick cathode electrolyte interphase (CEI) layer consisting of crystalline Li$_2$O (see the O 1$s$ XPS spectra in Fig. 4b and the Li 1$s$ XPS spectra in Supplementary Fig. 26) and amorphous matrix was formed on the cathode

surface at fully charged state. Li$_2$O is derived from the shuttling and co-reaction of soluble Li$^+$ into the passivation layer formed on the surface of the FeOCl-based positive electrode; while the amorphous matrix can be further assigned to alkyl carbonates/polycarbonates (see the C=O peaks at 532.5 eV and C–O peaks at 533.5 eV[34] in the O 1$s$ XPS spectra in Fig. 4b) as the oxidative decomposition products of DMA solvent. This thick and electrochemically inactive CEI layer with low strength (Young's modulus of 557 MPa, see the atomic force microscope (AFM) result in Supplementary Fig. 27) covers the cracked FeOCl (see the TEM images in Fig. 4a and Fe 2$p$ XPS spectra in Fig. 4b) surface, increasing the irreversibility of the battery system. As seen from the ex situ XRD patterns in Fig. 4c, with the increasing of charging voltages at 25 °C and 50 mA g$^{-1}$, the FeOCl peak was absent in the cell using the DMA-based electrolyte due to the continuous electrolyte decomposition at around 3.0 V (Supplementary Fig. 25), which is well consistent with the TEM and XPS analysis. Notably, it is seen that after the fluorination of DMA into FDMA and the introduction of HFE into the electrolyte, the thickness of CEI was reduced, meanwhile the structural integrity of FeOCl was maintained after cycling (Supplementary Fig. 28a, b). In the HGPE, a CEI layer with thickness as low as approximate 2 nm (Fig. 4d) and homogeneous FeF$_3$ species distriubtion (see FeF$_3$ peaks at 714.2 eV for Fe 2$p_{3/2}$ and 727.2 eV for Fe 2$p_{1/2}$[35] in Fe 2$p$ XPS spectra in Fig. 4e) was observed on the FeOCl-based electrode surface, corresponding to an enhanced strength of the cycled electrode surface (Young's modulus of 891 MPa, see the AFM result in Supplementary Fig. 29); while no Li$_2$O as the side reaction product of soluble Li$^+$ was detected (see the O 1$s$ XPS spectra in Fig. 4e and the Li 1$s$ XPS spectra in Supplementary Fig. 30). Furthermore, the ex situ XRD patterns (Fig. 4f) validate that in the presence of HGPE, the FeOCl cathode underwent a reversible transition from FeOCl to amorphous FeO accompanied by reversible Cl-ion storage without structural deterioration. The above cathode|electrolyte interphase analyses well supports the failure mechanism illustrated by Eqs. 3–5, and proves that the synergistic contribution of fluorinated solvents and the

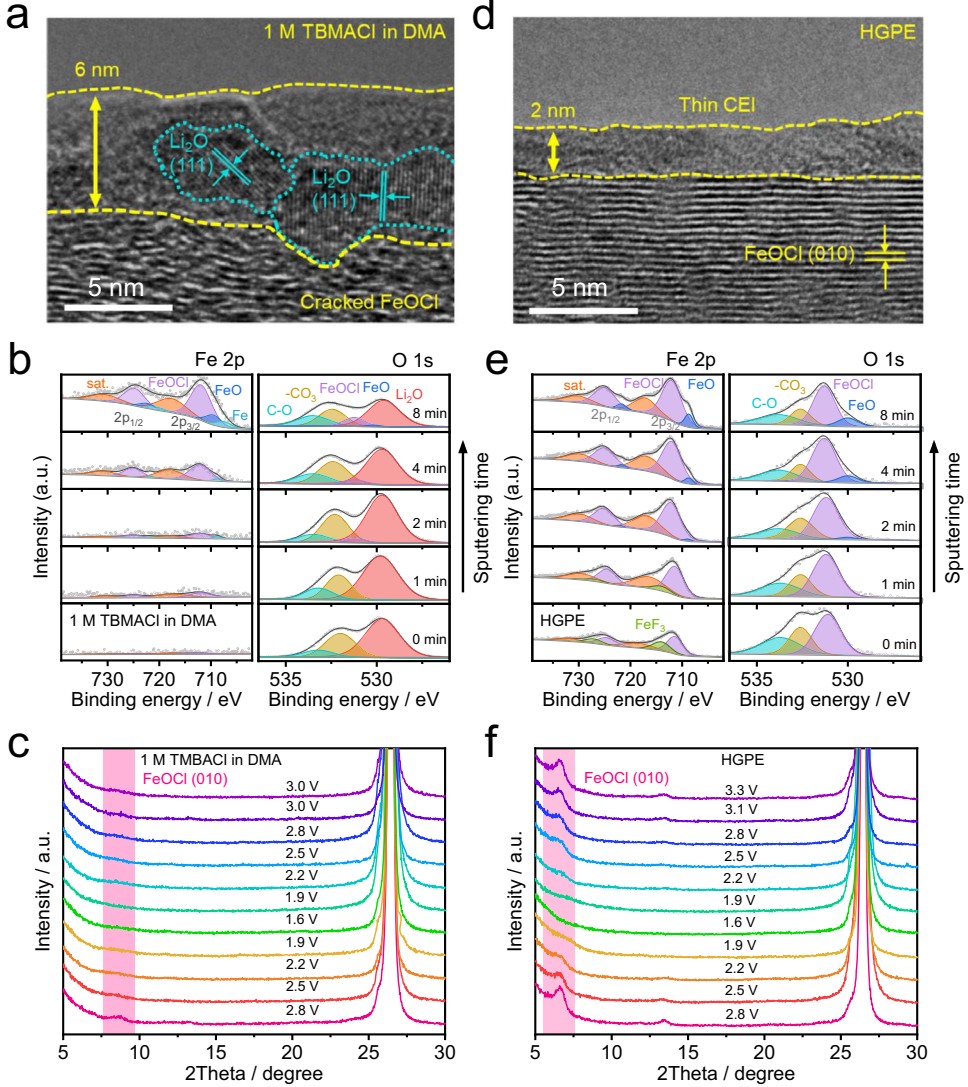

**Fig. 4 | Ex situ physicochemical characterizations of the FeOCl-based electrodes. a** TEM images of FeOCl electrodes sampled from Li‖FeOCl coin cells with 1 M TBMACl in DMA electrolyte after 3 cycles at 125 mA g⁻¹ and 25 °C. **b** Ex situ in-depth Fe 2*p* and O 1*s* XPS spectra of FeOCl cathodes in 1 M TBMACl in DMA electrolyte. **c** Ex situ XRD patterns of FeOCl-based electrodes cycled in Li metal coin cells with 1 M TBMACl in DMA electrolyte at 50 mA g⁻¹ and 25 °C. **d** TEM images of FeOCl electrodes sampled from Li‖FeOCl coin cells with Cl-ion-conducting HGPE after 3 cycles at 125 mA g⁻¹ and 25 °C. **e** Ex situ in-depth Fe 2*p* and O 1*s* XPS spectra of FeOCl cathodes in HGPE. **c, f** Ex situ XRD patterns of FeOCl-based electrodes cycled in Li metal coin cells with HGPE at 50 mA g⁻¹ and 25 °C.

polymer matrix can improve the compatibility between cathode and electrolyte.

Unlike Li-ion batteries in which the Li metal anode tends to generate dendrite structure during the stripping/plating process, the HIBs have been reported to be dendrite-free on the metallic anodes[1]. However, the interphase compatibility between the non-aqueous electrolytes and metallic anodes, in particular for the halide-ion-conducting passivation layer, is not adequately investigated in the literature. Here, the voltage variations of Li|Cl-ion-conducting electrolyte|LiCl@Li (where LiCl@Li indicates a LiCl-coated Li metal electrode) cells were measured at a constant current of 0.1 mA cm⁻² to investigate the interphase behavior at the Li metal anode. As shown in Fig. 5a, in the Li‖LiCl@Li cell with 1 M TBMACl in DMA and 1 M TBMACl in PC electrolytes, the polarization voltage fluctuated after 5 and 150 h, respectively. This indicates that in these electrolytes, the organic species-rich SEIs (see the high intensities of C–C/C–H peak at 284.8 eV and C–O/C=O peak at 286.6 eV in the C 1*s* XPS spectra[34] in Supplementary Fig. 31a, b) formed on the Li anode are unstable during cycling. When fluorinated FDMA was used as the electrolyte solvent,

the Li‖LiCl@Li cell exhibited generally increased overpotentials during the 650 h-cycling. Meanwhile, the LiF content increased (see LiF peak at 57.8 eV in the Li 1*s* XPS spectra[34] in Supplementary Fig. 31c, d) in the SEI, leading to improved interphase stability. As for the HGPE, the voltage hysteresis of the Li|HGPE|LiCl@Li cell was stabilized at around 100 mV without major oscillations (insets of Fig. 5a) during the 1000 h, indicating that the interaction force from the polymer matrix effectively hinders the movement of Cl ions toward the surface defects on the SEI, thus benefiting homogeneous Cl⁻ ionic flux to facilitate uniform and stable LiCl plating on the Li anode[36].

The fully discharged Li anodes disassembled from Li‖FeOCl cells after 3 cycles at 125 mA g⁻¹ were further characterized by ex situ field emission scanning electron microscopy (FE-SEM) measurements. It is worth noting that no Li dendrites were observed on the Li metal surfaces in all electrolytes, confirming the dendrite-free feature of metallic anodes in CIBs, which eliminates the safety hazards (e.g., short circuit and thermal runaway) originating from dendrite growth[12]. In the FE-SEM images of the electrodes it is possible to observe corrosion pits caused by Cl⁻ and cracks derived from the volume change (i.e., 56.5%

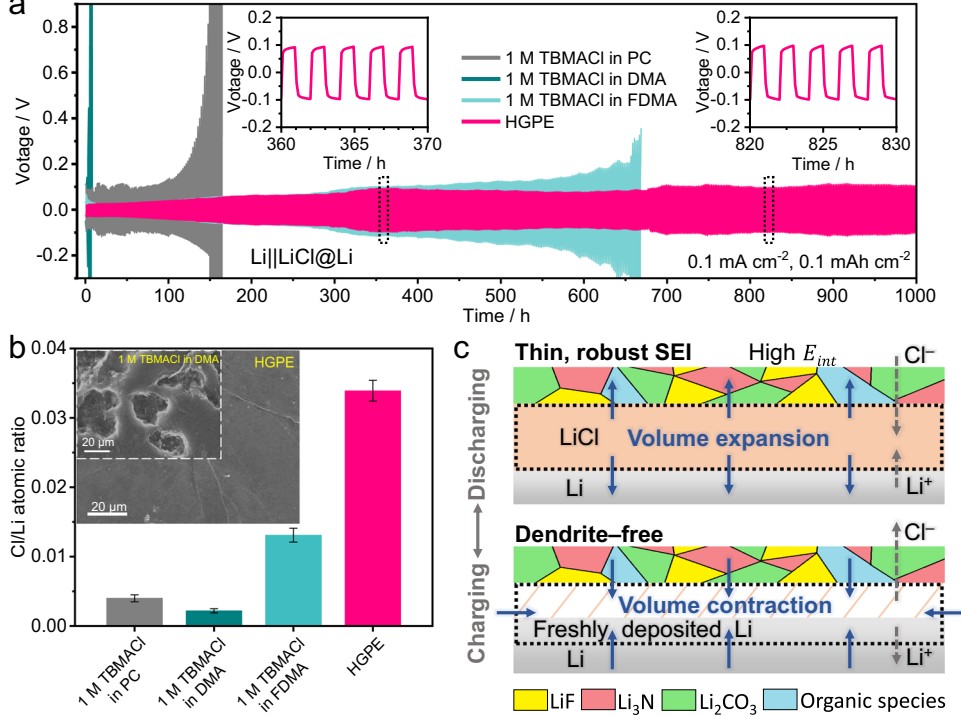

**Fig. 5 | Electrochemical and physicochemical characterizations of the Li|electrolyte interphase. a** Voltage profile of Li||LiCl@Li cells using different electrolytes at 0.1 mA cm⁻² with a cut-off capacity of 0.1 mAh cm⁻². Insets are enlarged portions of 360–370 h and 820–830 h for the Cl-ion-conducting HGPE-based cell. **b** Cl/Li atomic ratio derived from the ex situ XPS spectra and ex situ FE-SEM images (shown in insets) of Li metal surfaces obtained from Li||FeOCl cells with different electrolytes after 3 cycles at 125 mA g⁻¹ and 25 °C. Error bars represent standard error of the mean. **c** Schematic illustrations of dendrite-free deposition behavior on the Li|HGPE interphase.

from Li to LiCl⁶) on the Li metal surfaces cycled in 1 M TBMACl in PC (Supplementary Fig. 32a) and 1 M TBMACl in DMA electrolytes (inset of Fig. 5b). The Cl/Li atomic ratio on such a negative electrode was lower than 0.005 (Fig. 5b). This indicates that discharge product LiCl cannot homogeneously deposit on the Li metal in these electrolytes with high LiCl solubility (Supplementary Fig. 5). By contrast, the Cl/Li atomic ratio on the Li metal surface cycled in 1 M TBMACl in FDMA electrolyte increased to 0.013 (Fig. 5b) and the anode surface became more even (Supplementary Fig. 32b). For the Li|HGPE|FeOCl cell, the Li metal anode exhibited a smooth surface (inset of Fig. 5b) under the protection of SEI with a Cl/Li atomic ratio of 0.034 (Fig. 5b). This result is consistent with the intensity variation of LiCl peaks in Cl 2$p$ XPS spectra (Supplementary Fig. 31), confirming the uniform solid deposition of LiCl on the Li metal anode. This can be validated by DFT calculations. As shown in Supplementary Fig. 33, the binding energy between DMA and LiCl (−1.48 eV) is stronger than that of Li-LiCl (−1.29 eV). It indicates that LiCl prefers to dissolve in 1 M TBMACl in DMA electrolyte rather than deposits on the Li metal anode, which leads to its shuttling and triggers unwanted side reactions. For comparison, the binding energies of FDMA-LiCl and HFE-LiCl were calculated to be −1.21 eV and −1.19 eV, respectively, which are weaker than that of Li-LiCl (−1.29 eV). As a result, LiCl tends to agglomerate as the solid deposit on the Li metal anode in the electrolytes with fluorinated solvents. Additionally, the binding energy between PETEA monomer and LiCl (−1.73 eV) is stronger than that between LiCl and solvent molecules. Therefore, the functional groups in the polymer matrix Cl-ion-conducting HGPE further immobilize the LiCl molecules to suppress their diffusion through the solvents. Such homogeneous solid deposition of LiCl favour the improvement of the HGPE-based CIBs electrochemical energy storage performances.

The interfacial compatibility between the HGPE and metallic Li anode can be summarized as follows. As shown in Fig. 5c, a LiF-rich SEI

with high robustness is constructed on the anode surface, which inhibits the continuous electrolyte consumption but allows the transfer of Cl ions to enable a reversible LiCl deposition/dissolution. Moreover, LiF shows a relatively high interphase energy ($E_{int}$, 73.28 meV Å⁻²)[37] and exhibits low adhesion to anode surface[38]. Hence, this LiF-rich SEI allows the separation of the anode at the interphase to accommodate the volume change during the conversion of Li to LiCl (Supplementary Fig. 34 and Supplementary Note 7), thus preventing the continuous breakdown/reconstruction of SEI and suppressing the delamination of LiCl (Fig. 5c)[38].

## Testing of halide-ion battery chemistries in practical cell configuration

Single-layer pouch cells with a FeOCl mass loading at the positive electrode of about 4.0 mg cm⁻² were assembled to further evaluate the CIB performance in practical applications (Fig. 6a). The HGPE-based pouch cell delivers an initial discharge capacity of 210 mAh g⁻¹ (based on the mass of FeOCl) at 125 mA g⁻¹ and 25 °C and an specific energy of 275.5 Wh kg⁻¹ (based on the total mass of Li anode and FeOCl) and maintained a 79.5% capacity after 100 cycles, which is higher than the cell using 1 M TBMACl in DMA electrolyte (Fig. 6b), making it a promising cost-effective battery system for large-scale energy storage applications (Supplementary Note 8). Notably, when aging the fully charged cells at 130 °C, the open circuit potential of the pouch cell using 1 M TBMACl in DMA electrolyte suddenly dropped down after 10 min (Fig. 6c) due to a contact failure caused by electrolyte thermal decomposition (see the swelling and bulging of pouch cell shown in the inset of Fig. 6c, left panel). In sharp contrast, the HGPE-based pouch cell maintained a constant thickness (the inset of Fig. 6c, right panel) and steady open circuit voltage during the aging test (Fig. 6c), owing to the thermal stability of fluorinated solvents and polymer matrix. Moreover, the cell using 1 M TBMACl in DMA electrolyte lost

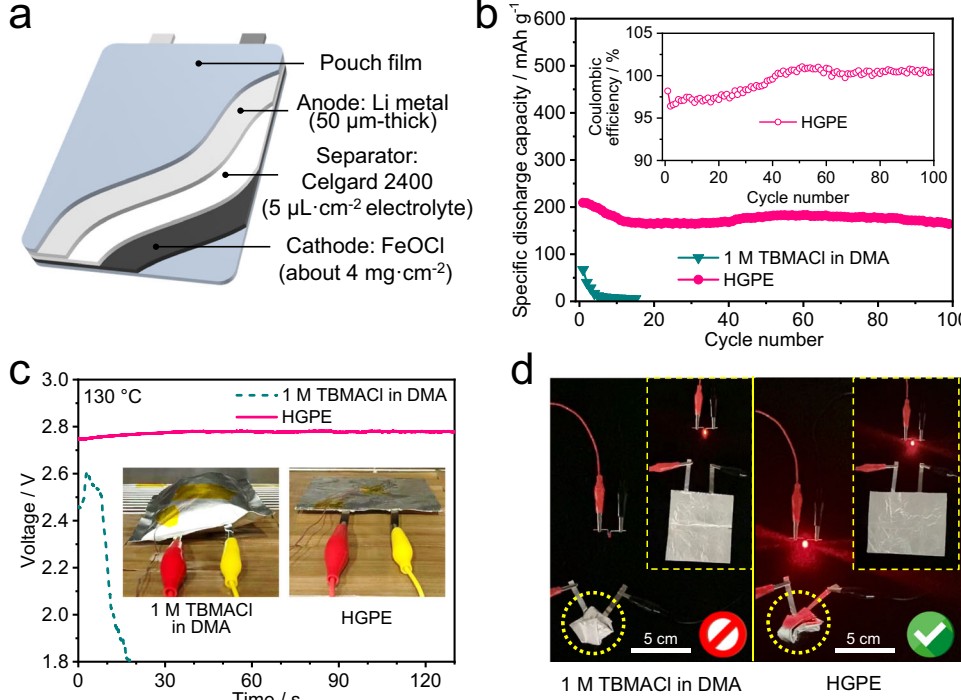

**Fig. 6 | Pouch cell characterizations of the quasi-all-solid-state CIB system.**
**a** Schematic illustration of Li||FeOCl pouch cells. **b** Cycling performance of Li||FeOCl pouch cells with 1 M TBMACl in DMA electrolyte and Cl-ion-conducting HGPE at 125 mA g⁻¹ and 25 °C. Inset is the Coulombic efficiency. **c** Open circuit voltage changes of fully charged Li||FeOCl pouch cells with 1 M TBMACl in DMA electrolyte (left) and CGPE (right) under 130 °C during the aging time. Optical images of the pouch cells after the aging test are shown in the insets. **d** Optical images of red LEDs powered by pouch cells with different electrolytes before (shown in the insets) and after severe shape deformation.

power supply in the bending conditions (left panel in Fig. 6d and Supplementary Movie 3); whereas the Li|HGPE|FeOCl pouch cell showed better flexibility (i.e., consistently powering a light-emitting diode (LED) under bending conditions (right panel in Fig. 6d and Supplementary Movie 4)). This validates that the electrode|HGPE interphases can keep tight adhesion under severe shape deformations. All these endow a reliable operation of this quasi-solid-state CIB under abuse conditions.

The HGPE concept can also be applied to other halide-ion shuttling chemistry. The generality of HGPE for the halide-ion shuttling batteries has been further demonstrated by FIB (Li||BiOF cell, Supplementary Fig. 35) and BIB (Li||BiOBr cell, Supplementary Fig. 36). The quasi-solid-state F-and Br–ion–conducting HGPEs were prepared through the same procedure as the Cl-ion-conducting HGPE except for replacing the chloride salt with equimolar concentration (1 M) tetrabutyl ammonium fluoride (TBAF) or tributyl methyl ammonium bromide (TBMABr), respectively. The F-and Br–ion–conducting HGPEs exhibited a morphology as translucent free-standing gels with high ionic conductivity of 4.52 or 2.42 mS cm⁻¹ at 25 °C, respectively (Supplementary Fig. 35a and Supplementary Fig. 36a). When paring with BiOF cathode (Supplementary Fig. 35b) or BiOBr cathode (Supplementary Fig. 36b) and 50 μm-thick Li metal anodes, the coin cells using HGPEs delivered initial discharge capacities of 122.2 mAh g⁻¹ for FIB (Supplementary Fig. 35c) and 75.7 mAh g⁻¹ for BIB (Supplementary Fig. 36c) at 100 mA g⁻¹ and 25 °C (based on the mass of the active material in the positive electrode). These cells maintained capacities of 60.5 mAh g⁻¹ (Supplementary Fig. 35d) and 33.7 mAh g⁻¹ (Supplementary Fig. 36d) after 50 cycles, respectively, which are far better than the cells using DMA-based electrolytes (Supplementary Fig. 35c, d, and Supplementary Fig. 36c, d). Bi 4$f$ and O 1$s$ XPS measurements (Supplementary Fig. 35e, f, Supplementary Fig. 36e, f and Supplementary Note 9) of the positive electrodes prove the reversible conversions between bismuth oxyhalides and Bi₂O₃/Bi during the discharge/charge

of F and Br ions in cathodes for HGPE-based FIBs and BIBs, respectively. Moreover, the ex situ XPS measurements of the FIB and BIB electrolytes indicate a lower Li and Bi elemental concentrations compared with the cells using DMA-based electrolytes (Supplementary Fig. 35f and Supplementary Fig. 36f). Moreover, the bismuth oxyhalides cathodes (Supplementary Fig. 37 and Supplementary Note 10) and the Li metal anodes (Supplementary Fig. 38 and Supplementary Note 11) show high reversibility in symmetric cells with HGPE, respectively. This demonstrates that the HGPEs efficiently restrain the dissolution of discharge products and cathode materials in FIBs and BIBs, thus enabling a highly reversible discharge/charge of F and Br ions.

In summary, we propose an effective electrolyte design strategy to develop quasi-solid-state non-aqueous halide ion batteries. Accordingly, a combined utilization of fluorinated low-polarity solvents paired with a gelation treatment was identified and rationalized as the optimal choice for quasi-solid-state electrolyte preparation. Using this design, a gel polymer electrolyte was facilely fabricated in situ by gelling a liquid electrolyte based on binary fluorinated solvents, which exhibited high bulk ionic conductivity (2.64 mS cm⁻¹ at 25 °C), high anodic stability (4.0 V vs. Li⁺/Li), non-flammability and resistance against liquid leakage. Applying this gel electrolyte, we developed a room-temperature Cl-ion full cell using FeOCl-based positive electrode and Li metal negative electrode and a Li metal anode, in which the dissolution of redox products and electrolyte oxidation issues were hindered by the contribution of fluorinated solvents and the polymer matrix in the electrolyte. Consequently, a quasi-solid-state Li||FeOCl coin cell delivered an initial discharge capacity of 230 mAh g⁻¹ at 125 mA g⁻¹ (based on the mass of FeOCl in the positive electrode) with remarkable cycling stability and safety under abuse conditions. The initial specific energy of the Li||FeOCl battery system (i.e., 275.5 Wh kg⁻¹ calculated for the 1ˢᵗ discharge and based on the total mass of Li metal and FeOCl as the positive and

negative electrodes, respectively) is well-positioned in the state-of-the-art literature of Cl-ion, Na-ion, K-ion and multivalent metal-ion batteries. Furthermore, the gel polymer electrolyte has been successfully extended to develop highly reversible F-and Br-ion batteries.

## Methods

### Preparation and characterization of the HGPEs

Pentaerythritol tetraacrylate (PETEA, Sigma-Aldrich, 98 %) and 2,2′−azobis(2−methylpropionitrile) (AIBN, Aladdin, 99 %) were stored at −5 °C before use. Tributyl methyl ammonium chloride (TBMACl, Adamas-beta, 99 %), tetrabutyl ammonium fluoride (TBAF, Aladdin, 98.0 %), tributyl methyl ammonium bromide (TBMABr, Aladdin, 98.0 %), N, N-dimethylacetamide (DMA, Aladdin, 99.8 %), 2,2,2−trifluoro−N, N-dimethylacetamide (FDMA, TCI, 98 %) and 1,1,2,2−tetrafluoroethyl 2,2,2−trifluoroethyl ether (HFE, DoDoChem, 99.95 %) and propylene carbonate (PC, DoDoChem, 99.95 %) were directly used without further purification. To prepare the HGPEs, 1.5 wt% PETEA monomer and 0.1 wt% AIBN initiator were dissolved in liquid electrolytes consisting of 1 M quaternary ammonium halide salt (TBMACl, TBAF, or TBMABr) in a mixture of FDMA: HFE (1: 1 by volume) as precursor solutions. Then the precursor solutions were in situ polymerized at 60 °C for 30 min on a hotplate in the Ar-filled glovebox ($H_2O$ and $O_2$ < 0.1 ppm) to obtain the translucent quasi-solid-state HGPEs.

$Cl_2$ solubility was quantified by bubbling the $Cl_2$ gas (99.999%) into electrolytes until saturated, then titrating the solutions through an indirect iodimetry method[39]. Fe solubility in the electrolyte was measured after immersing 1 g FeOCl in the electrolyte for 7 days. The element concentration was measured by an inductively coupled plasma optical emission spectrometry (ICP-OES, Arcos II MV, SPECTRO). The dissolution of FeOCl and LiCl in an ionic liquid electrolyte (i.e. 0.5 M TBMACl in $PP_{14}TFSI$ (1−Butyl−1−methylpiperidinium bis(trifluoromethylsulfonyl)imide)) was test as follow. 0.5 mg (excess amount) of LiCl or FeOCl was immersed in such an ionic liquid electrolyte for 7 days. The supernatant was diluted by ultrapure water and measured by the ICP-OES. The Fourier transform infrared spectra (FTIR) spectra of the PETEA monomer and the polymer matrix of HGPE were measured with a Bruker Vertex70 instrument. To obtain the polymer matrix from the HGPE, HGPE was firstly cut into pieces (<1 mm) and washed with acetone. The Raman spectroscopy measurements was conducted by a Micro-laser confocal Raman spectrometer (Horiba LabRAM HR800, France) at 25 °C with a 532 nm laser. In the combustion test, 1 g electrolyte samples were poured into a C2032 coin cell shell, and then optical photographs and movies were recorded after the samples were ignited. The ionic conductivities of the electrolyte samples were measured by electrochemical impedance spectroscopy measurements (EIS) with a frequency range of $10^5$ Hz to 1 Hz with an alternating current amplitude of 5 mV on a VMP3 multi-channel electrochemical station (Bio Logic Science Instruments, France) with platinum black block electrodes (Leici DJS−1 C, Shanghai). Before the conductivity measurements, the testing cells were maintained at each test temperature (20 to 90 °C) for at least 30 min to reach thermal equilibrium. The electrochemical stabilities of the electrolytes were studied by linear sweep voltammetry (LSV) in coin cells using titanium foil (99.99%, with a thickness of 100 μm, MTI Corporation, China) as the working electrode and Li foil (99.95%, with a thickness of 50 μm, China Energy Lithium Co., Ltd.) as both counter and reference electrode at a scanning rate of 10 mV s⁻¹ using the VMP3 electrochemical station at 25 °C.

### Battery assembly and characterization

CR2032 coin cell was used, and the batteries were assembled in the Ar-filled glovebox ($H_2O$ and $O_2$ < 0.1 ppm). The FeOCl cathode material was prepared by heat treatment of $FeCl_3·6H_2O$ (Aladdin, 99%) at 180 °C for 10 h under vacuum, followed by washing with acetone (Aladdin,

99.5%) and drying at 120 °C in a vacuum oven[20]. The BiOF cathode material was prepared through a fluorination treatment of $Bi_2O_3$ (Aladdin, 99.99 %) by a saturated aqueous solution of $NH_4F$ (Aladdin, 98.0%). The BiOBr cathode material was prepared through an ion exchange method of precipitating the mixture of 1 M $Bi(NO_3)_3·5H_2O$ (Aladdin, 99.99%) and 1 M KBr (Aladdin, 99.99%) aqueous solution. The cathodes of HIBs were prepared by a slurry-coating method. First, a slurry consisting of 70 wt% metal oxyhalide as active material, 20 wt% Super-P (40 nm, TIMCAL) as a conductive agent, and 10 wt% polyvinylidene fluoride (PVDF, Macklin, AR) as the binder in anhydrous 1-methyl-2-pyrrolidinone (NMP, Aladdin, 99.5%) was cast onto a graphite foil (0.02 mm, Jinglong Carbon Technology Co., Ltd) as the current collector, and then dried at 120 °C for 24 h in a vacuum oven. The mass loading of metal oxyhalides on the cathode was around 1.0 mg cm⁻². Celgard 2400 separator was applied and Li metal foil was used as the anode. Two pieces of titanium foils were placed between electrodes and gaskets to prevent the corrosion of halide ions to stainless-steel gaskets. A precursor solution containing 1.5 wt% PETEA and 0.1 wt% AIBN dissolved in 1 M quaternary ammonium halide in a mixture of FDMA: HFE (1: 1 by volume) electrolyte was injected into the separator and filled into the cells. The electrolyte: cathode active material ratio in each cell was set as about 40 μL mg⁻¹. The assembled cells were aged at 25 °C for 2 h to ensure the precursor solution well-wetted the electrodes, and then the cells were heated at 60 °C for 1 h to ensure an in situ polymerization of PETEA to get the HGPE-based cells.

The CV measurements of Li‖FeOCl coin cells were recorded from 1.6 to 3.3 V at a scanning rate of 0.1 mV s⁻¹ using the VMP3 electrochemical station at 25 °C. The discharged/charged voltage was set at 1.6 V/3.3 V for Li ‖FeOCl coin cells and 1.6 V/2.8 V for Li‖bismuth oxyhalide coin cells for galvanostatic cycling test on a NEWARE battery testing system at 25 °C. A cut-off time of 2 h was set in the charging process to avoid electrolyte decomposition. The self-discharge of Li‖ FeOCl cells was investigated by recording the static voltage of fully charged cells for 100 h at 25 °C. To fabricate the Cl-ion-shuttling Li‖Li symmetric cells, Li‖FeOCl cells with 1.0 mg cm⁻² FeOCl loading were fully discharged at 125 mA g⁻¹, then the cells were re-assembled by replacing the FeOCl cathode with a fresh Li anode. The Li‖LiCl@Li(LiCl discharge product on Li metal) cells were cycled at a capacity of 0.1 mAh cm⁻² under 0.1 mA cm⁻². EIS measurements were carried out at a frequency of $10^{-1}$ to $10^6$ Hz by applying a disturbance amplitude of 5 mV on the VMP3 electrochemical station at 25 °C. The cells after designated cycling tests were dissembled in an argon-filled glove box, and the electrodes were repeatedly rinsed with dimethyl carbonate (DMC, in which the LiCl solubility was as low as $1.35 × 10^{-4}$ M) before post-mortem analysis. For the ex situ test of electrodes, coin cells after cycles were dissembled in the Ar-filled glovebox ($H_2O$ and $O_2$ < 0.1 ppm). The ex situ in-depth XPS was carried out on a PHI 5000 Versa Probe II spectrometer using a monochromatic Al Kα X-ray source at 1486.6 eV with argon-filled transfer box. The sputtering rate is about 3 nm min⁻¹. Ex situ TEM was conducted by a Tecnai $G^2$ F30 TEM instrument. Ex situ FE-SEM and EDS were conducted by a HITACHI SU8010 FE-SEM. The Cl/Li atomic ratio derived from the EDS spectra. Ex situ Surface analysis was carried out on a Bruker Dimension Icon AFM. The ex situ XRD patterns were characterized on a Rigaku D max 2500 diffractometer with Cu Kα radiation (λ = 1.5418 Å). Electrodes obtained from cycled coin cells in the Ar-filled glovebox ($H_2O$ and $O_2$ < 0.1 ppm) were protected by the Kapton tape for the XRD test.

Li‖HGPE‖FeOCl pouch cells were assembled in the Ar-filled glovebox ($H_2O$ and $O_2$ < 0.1 ppm) following a similar method. The FeOCl mass loading was about 4.0 mg cm⁻² for pouch cells. A nickel strip and an Al strip were anchored to Li foil and titanium foil current collector as electrode tabs, respectively. The electrodes (area of 20 cm²) were laminated together with the separator to construct the battery core in the aluminum-plastic film packages (5 cm), followed by injecting the electrolyte precursor solution (about 5 μL cm⁻²) into the packages and

sealing the packages under a vacuum. The assembled pouch cells (thickness of 0.5 mm) were aged for 2 h to ensure the precursor solution wetted the electrodes sufficiently. Subsequently, the cells were heated at 60 °C for 0.5 h to get the quasi-solid-state pouch cells. The high-temperature stability test was performed by detecting the voltage change of the pouch cells at 130 °C with a data logger (LR8431-30, HIOKI) in a high-temperature battery-explosion-insulation testing machine (DMS-9987, Demaisheng Measurement and Control Equipment Co., Ltd, Shenzhen).

The MD simulations were run using LAMMPS[40]. The systems are set up initially by using PACKMOL[41] and Moltemplate (http://www.moltemplate.org/). Periodic boxes were used here. The force-fields parameters of solvents and salt are taken from OPLS[42] and assigned with RESP charges[43,44]. An LJ cutoff of 10 Å and a particle-particle particle–mesh solver[45] for long-range Coulombic interactions were also employed. The Velocity–Verlet algorithm was applied to integrate the equations of motion with a time step equaled to 1fs. First, Langevin dynamics were performed at 500 K for 1 ns and then NPT runs at 298 K were performed for 10 ns to ensure that the equilibrium salt dissociation had been reached. Then, the NVT runs were 20 ns long at 298 K. The last 10 ns trajectory was used to obtain the structure of the electrolyte. Visualization of the electrolyte structures is made by using VESTA[46] and VMD[47]. The geometries of isolate LiCl, FDMA, DMA, PC, PETEA, and HFE were optimized under the framework of DFT with PBE0[48] functional and def2SVP[49] basis set. 100 possible absorb structures of each complex were generated and searched using a molecules program (Tian Lu, Molclus program, Version 1.9.9.2, accessed Aug. 4, 2021, http://www.keinsci.com/research/molclus.html). These clusters were optimized using the GFN–xTB program[50] first. After the preliminary structural optimization, 10 clusters with lower energy were selected and then further optimized with dispersion corrected density functional theory (DFT–D3) at the PBE0–D3/def2–SVP level[48,49] using the Gaussian program. Dipole moments (μ) of optimized molecule structures were obtained from the Multiwfn program[44]. Electrostatic potential (ESP) was obtained from the Gaussian view program. The binding energy of the complex was calculated from the formula: "$E(binding) = E(A+B) - E(A) - E(B)$".

## Data availability

The data that support the findings of this study are available from the corresponding author upon reasonable request.

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

## Acknowledgements

G.W. would like to acknowledge the support from the Australian Research Council (ARC) Discovery Projects (DP200101249 and DP210101389) and the ARC Research Hub for Integrated Energy Storage Solutions (IH180100020). B.L. would like to acknowledge the support funded by the National Nature Science Foundation of China (No. 51872157), Shenzhen Key Laboratory (ZDSYS201707271615073), and Guangdong Technical Plan Project (No. 2017B090907005).

## Author contributions

D.Z., B.L., C.W. and G.W. conceived and designed this work. D.Z. and X.Y. performed the experiments and wrote the manuscript. B.Z. conducted molecular dynamics simulations. Y.T. performed the DFT calculations. Y.W., B.Z., Z.F., H.L., and F.K. discussed the results and participated in the preparation of the paper.

## Competing interests

The authors declare no competing interests.
