## [Peer Review File · Nature Communications]

Electrolyte design principles for developing quasi–solid–state rechargeable halide–ion batteriesEditorial Note: This manuscript has been previously reviewed at another journal that is not operating a transparent peer review scheme. This document only contains reviewer comments and rebuttal letters for versions considered at *Nature Communications* .

REVIEWER COMMENTS

Reviewer #1 (Remarks to the Author):

The manuscript has been well revised according to the reviewers' comments. A minor revision is suggested by considering the following comments.

1. In the supplementary Table 3, it is interesting that the FeOCl exhibits a high solubility (2.3 mol L⁻¹) in the 0.5 M TBMAcI in PP14TFSI solution. This did not occur in the previously reported well-dried electrolyte of 0.5 M PP14Cl in PP14TFSI. Is there any residue left in the solution after the immersion test? If there was residue, what's the phase structure of it?
2. It was claimed that the use of the hybrid solvent avoided the self-discharge and dissolution issues which have caused the decay of FeOCl cathode. However, the FeOCl cathode in the electrolyte using the hybrid solvent still exhibited an evident capacity decay. Only a capacity of ~120 mAh/g remained at around the 30th cycle, which was much lower than that of the previously reported FeOCl/CMK-3 cathode in the ionic liquid electrolyte (ACS Energy Lett. 2017, 2, 2341). This capacity decay should be explained.
3. For the Ar-sputtering XPS test of FeOCl, it is clear that the Fe 2p peaks shift toward lower binding energy by 4 min Ar-sputtering (Figure R2 in the first response), indicative of a chemical reduction of iron. Note that this chemical reduction may be more severe for the recharged FeOCl sample.
4. For the mechanisms of fluoride ion or boride ion shuttle, the halide-ion transfer at the anode side was not proved. For example, was LiF formed after discharge and could it be reduced to lithium metal after recharge in the fluoride electrolyte?

Reviewer #2 (Remarks to the Author):

Although the authors have used several "for the first time" to emphasize the novelty of this article, I still insist my opinion that this work shows the limited scientific significance. On the other hand, it is undoubted that the authors achieved significant improvements in comparison with previous reports. Furthermore, the authors also deal with my previous concerns. After hesitation, this reviewer would like to recommend it for publication on NC.

Response to Reviewers' Comments

Reviewer #1 (Remarks to the Author):

The manuscript has been well revised according to the reviewers' comments. A minor revision is suggested by considering the following comments.

1. In the supplementary Table 3, it is interesting that the FeOCl exhibits a high solubility (2.3 mol L⁻¹) in the 0.5 M TBMAcI in PP14TFSI solution. This did not occur in the previously reported well-dried electrolyte of 0.5 M PP14Cl in PP14TFSI. Is there any residue left in the solution after the immersion test? If there was residue, what's the phase structure of it?

Reply: Thanks for the Reviewer's positive evaluation on the quality of our manuscript and the valuable comments. In our immersion test, 0.2g of pristine FeOCl was immersed in 0.5 M TBMAcI in PP₁₄TFSI electrolyte (with a moisture content of 34.6 ppm determined by Karl Fischer titration method) and stirred for 72 h. The residue was washed by acetone and dried under vacuum overnight. As shown in **Figure R1a**, the FeOCl (001) peak intensity of residue was weakened compared with the pristine FeOCl (PDF#73-2229), indicating a partial structural deterioration of layered FeOCl after the immersion test accompanied by a formation of amorphous by-product. The Fe 2p XPS spectra of the residue in **Figure R1b** further shows an increased Fe₂O₃ content while the energy-dispersive X-ray spectroscopy (EDS) element analysis in **Table R1** exhibits a decreased Cl content on the residue surface compared with the pristine FeOCl, confirming the side reaction proposed in our Manuscript:

Figure R1 **a** XRD patterns and **b** Fe 2p XPS spectra of pristine FeOCl and the residue after immersion test.

Table R1 EDS element analysis of pristine FeOCl and the residue after immersion test.

Element	Atomic% in pristine FeOCl	Atomic% in residue
O	37.3	39.6
Cl	30.2	28.4
Fe	32.5	32.0

2. For the Ar-sputtering XPS test of FeOCl, it is clear that the Fe 2p peaks shift toward lower binding energy by 4 min Ar-sputtering (Figure R2 in the first response), indicative of a chemical reduction of iron. Note that this chemical reduction may be more severe for the recharged FeOCl sample.

Reply: Thanks for the Reviewer's important comment. We acknowledge the Reviewer's opinion that the Ar⁺ sputtering process possibly cause unwanted reduction reaction during the

in-depth XPS analysis (*ACS Energy Lett.* **7**, 3270 (2021)). As mentioned by the Reviewer, for the FeOCl powder, the Fe 2p peaks slightly shifted toward lower binding energy by 4 min Ar⁺ sputtering. Actually, this peak shift was mainly originated from the removal of surficial Fe₂O₃ rather than a chemical reduction of iron by Ar⁺ sputtering. The Fe 2p_{1/2} and Fe 2p_{3/2} peaks of FeOCl maintained fixed positions during the subsequent 4–8 min Ar⁺ sputtering (**Figure R2**), implying the impact of Ar⁺ reduction reaction on the in-depth XPS results is negligible.

Figure R2. In-depth Fe 2p XPS spectra of the FeOCl powder.

3. It was claimed that the use of the hybrid solvent avoided the self-discharge and dissolution issues which have caused the decay of FeOCl cathode. However, the FeOCl cathode in the electrolyte using the hybrid solvent still exhibited an evident capacity decay. Only a capacity of ~120 mAh/g remained at around the 30th cycle, which was much lower than that of the previously reported FeOCl/CMK-3 cathode in the ionic liquid electrolyte (*ACS Energy Lett.* **2017**, **2**, 2341). This capacity decay should be explained.

Reply: In our research, the FeOCl active material was uncoated, while in *ACS Energy Lett.* **2**, 2341 (2017), FeOCl was confined in CMK-3 mesoporous carbon. This leads to remarkable difference in cycle stability, since the carbon matrix is expected to enhance the electronic conductivity and suppress the dissolution issues of FeOCl. As shown in **Figure R3**, FeOCl/CMK-3 sample with a carbon content of 46.6 wt% was prepared according to *ACS Energy Lett.* **2**, 2341 (2017), and applied as cathode material in chloride-ion cell using 1 M TBMAcI in FDAM: HFE hybrid solvents in the electrolyte. It is seen that the cycling performance of Li||FeOCl/CMK-3 cell shows much higher capacity retention (74.4 % after 30 cycles) compared with the Li||pristine FeOCl cell (60.9 % after 30 cycles), which is comparable

to the cell using ionic liquid electrolyte reported on *ACS Energy Lett.* **2**, 2341 (2017) (*i.e.*, ~80.2% after 30 cycles). This confirms the effect of hybrid solvent on electrolyte optimization. We have cited this paper as the Reference 20 on Page 32 in the Revised Manuscript.

Figure R3 Cycling performance of the Li||FeOCl/CMK-3 cell and Li||FeOCl cell with 1 M TBMAcI in FDMA:HFE hybrid solvent electrolytes at 0.5 C (based on the mass of FeOCl).

4. For the mechanisms of fluoride ion or boride ion shuttle, the halide-ion transfer at the anode side was not proved. For example, was LiF formed after discharge and could it be reduced to lithium metal after recharge in the fluoride electrolyte?

Reply: Thanks for the Reviewer's comments. To verify the reversibility of halide-ion transfer on the Li metal anode during cycling, experiments were designed as follow. First, fluoride-ion battery (FIB, Li||BiOF cell) and bromide-ion battery (BIB, Li||BiOBr cell) were assembled and cycled for 5 cycles to form the stable SEI on the Li anodes. Then cells were disassembled and small parts of Li metal anodes at charged state were cut off for SEM characterization. The rest of Li metal anodes were re-assembled into the halide-ion cells, and small parts from the fully discharged Li metal anodes were tested by SEM. After that, the rest of Li metal anodes were re-assembled in the halide-ion cells and re-charged for SEM characterization. As shown in Supplementary Figure 31a-c, in FIB, the F element ratio on the Li anode increased from 29.7 % in the charged state to 50.8 % in the discharged state, and then dropped back to 31.2 %, accompanied by a reversible change of surface roughness. The similar tendency was also observed in BIB (Supplementary Figure 31d-f). These results confirm a highly reversible halide-ion transfer on the anode side. We have added above discussion as Supplementary Figure 31 on Page 31 in the Revised Supplementary Information.

Supplementary Figure 31. FE-SEM images of Li metal anodes cycled in **a–c** FIBs and **d–f** BIBs full cells at different state of charge. EDS element analyses are shown in insets. Scale bars: **10 μm .**

Reviewer #2 (Remarks to the Author):

Although the authors have used several “for the first time” to emphasize the novelty of this article, I still insist my opinion that this work shows the limited scientific significance. On the other hand, it is undoubted that the authors achieved significant improvements in comparison with previous reports. Furthermore, the authors also deal with my previous concerns. After hesitation, this reviewer would like to recommend it for publication on NC.

Reply: Thanks for the reviewer’s positive evaluation on the quality of this manuscript.

REVIEWERS' COMMENTS

Reviewer #1 (Remarks to the Author):

The manuscript is suggested to be accepted for publication in Nature Communications.